

# Seafloor marine heatwaves outpace surface events in future on the northwest European shelf

Robert J. Wilson[1], Yuri Artioli[1], Giovanni Galli[1,2], James Harle[3], Jason Holt[4], Ana M. Queirós[1,5], and Sarah Wakelin[4]

[1]Plymouth Marine Laboratory, Prospect Place, Plymouth PL1 3DH, United Kingdom
[2]National Institute of Oceanography and Applied Geophysics (OGS), Section of Oceanography, via Beirut 2, Trieste, 34014, Italy
[3]National Oceanography Centre (NOC), European Way, Southampton, SO14 3ZH,
[4]UK National Oceanography Centre (NOC), Joseph Proudman Building, 6 Brownlow Street, Liverpool, L3 5DA, UK
[5]Faculty of Environment, Science and Economy, University of Exeter, UK

*Correspondence*: Robert J Wilson (rwi@pml.ac.uk)

**Abstract.** Marine heatwaves are increasingly frequent across the world's oceans. As a result, there are growing impacts on marine ecosystems due to temperatures exceeding the thermal niche and historical exposure of many species. Anticipating the future frequency and severity of marine heatwaves is necessary. Here we provide the first projections of future marine heatwaves for the sea surface and seafloor across the northwest European Shelf, which is a critically important marine ecosystem. We use an ensemble of five dynamically downscaled hydrodynamic models under the high emissions scenario RCP 8.5. Heatwaves were defined as events lasting at least 5 days where temperatures exceed the 90th percentile of a historical baseline period. The frequency of marine heatwaves at the surface and seafloor is projected to increase significantly during the 21st century under RCP 8.5, with most of the year projected to be in heat wave conditions by the end of the century. Critically, we find that marine heatwaves are projected to increase in frequency to a greater extent at the seafloor compared with the sea surface due to their lower levels of natural temperature variation. Similarly, we find that the severity of summer heatwaves at the surface is projected to be lower than that of heatwaves during the rest of the year, due to lower climatological variation in temperature outside the summer. The impacts of marine heatwaves on shelf seas are therefore likely to be much more complex than anticipated heretofore, when taking a view beyond the ocean surface and summer.

## 1 Introduction

Marine heatwaves are defined as prolonged periods where the sea is anomalously warm (Oliver et al. 2021). Since Pearce et al. (2011) first used the term, there has been a rapid growth in our understanding of their causes and how they impact ecosystems. In recent decades, rising temperatures have caused marine heatwaves to become more and more frequent around the world (Oliver et al. 2019, Oliver et al. 2018a, Marin et al. 2021, Oliver et al., 2018b, Scannell et al., 2016, Darmaraki et



al. 2019, Yao et al., 2023). They have multiple causes, ranging from the occurrence of atmospheric heatwaves, changes in wind speeds, and variations in ocean mixing (sen Gupta et al., 2020, Amaya et al., 2020).

Marine heatwaves have largely negative impacts on marine ecosystems (Smith et al. 2022). Negative impacts on seaweeds (Thomsen et al. 2019, Gurgel et al. 2020, Arafeh-Dalmau et al., 2019, Filbee-Dexter et al. 2020), seagrasses (Arias-Ortz et al. 2018, Styrdom et al. 2020), seabirds (Jones et al. 2018, Piatt et al., 2020), coral reefs (Leggat et al. 2019, Le Nohaïc et al. 2017), crustaceans (Chandrapavan et al. 2019), fish (Wild et al. 2019, Roberts et al. 2019), and plankton (Brodeur et al. 2019, Nielsen et al., 2021) have been recorded, among others. However, the impacts of marine heatwaves are complex and

vary across ecosystems (Smale et al. 2017) and it remains unclear to what extent ecosystems can adapt to the continued rise in their frequency (Pershing et al. 2018).

The impacts of marine heatwaves on the sea surface and seafloor are likely to be distinct. Pelagic species are typically advected or can swim across large regions and can therefore acclimate or reorganize in response to climate change relatively quickly. In contrast, benthic and demersal species tend to have limited mobility, such that tracking optimal thermal habitat

e.g. though the deepening of assemblages is seen as more challenging. It is thus more likely that negative impacts are observed (e.g. Queiros et al. 2021, 2023), especially in species with slow growth rates which slow the ability to respond to disturbance (Ceccherelli et al. 2007). Effective management of these species and habitats thus necessitates that projections of future marine heatwaves at both the ocean surface and the seafloor are considered. While most of the marine heat wave assessments to date have been looking at sea surface temperature data, recent studies suggest that sub-surface heat waves can

be equally relevant (He et al. 2023, Sun et al. 2023).

A large body of work has used global climate models to project how future marine heatwaves are expected to increase this century (Azarian et al. 2023, Cheng et al. 2023, Oliver et al. 2019. Plecha et al. 2019, Qiu et al. 2021, Rosselló et al. 2023, Sun et al. 2023, Xue et al. 2023). However, global models are substantially limited in their ability to project marine heatwaves at both the sea surface and seafloor on shelf seas. Global models typically have coarse spatial resolution, which

can result in poor representation of ocean shelf processes, and the representation of critical processes such as riverine inputs which affect key processes in shelf environments are poor (Holt et al. 2017, Kay et al. 2023). In combination, this results in poor representation of processes such as seasonal thermal stratification of the water column, resulting in modelling projections which do not provide sufficiently credible representation of coastal processes. Finally, marine heatwave calculations require daily temperature data (Hobday et al. 2016). This is available for the sea surface for many global climate

models (Wilson et al. 2024), however daily near-bottom temperatures are not available in those, including the models used in the global climate change modelling standard, the Coupled Model Intercomparison Project Phases 5 and 6 (CMIP5 and 6). This leads to an unavailability of reliable ocean modelling projections from which to derive marine heatwave estimations to inform marine science, policy and industry development operating at international, national, and sub-national level.

The limitations of global climate models on continental shelves can largely be resolved using dynamical downscaling (e.g.

Drenkard et al 2021), whereby a regional model with higher horizontal and vertical resolution is used to project future changes in ocean physics and biogeochemistry (e.g., Butenschön et al. 2016). Through this approach, a regional model is forced at the boundaries, i.e., at the air-surface boundary and the geographic limits of the regional model domain, by the





outputs of a global model, offering a higher resolution of processes within the region of interest, whilst being consistent with the broader processes simulated in the global model. Here we apply this approach to understand future marine heatwaves on

the northwest European shelf.

The northwest European shelf is an important marine ecosystem that provides ecological, cultural, and economic services to many countries (Culhane et al., 2020). It is one of the world's most heavily fished regions (Halpern et al. 2008) and has the world's largest and fastest growing offshore wind industry. Understanding how climate change will impact the region is thus critical to enable effective spatial and fisheries management into the future (Queirós et al. 2021).

In the north-west European shelf, a suite of dynamically downscaled projections exists and have been used to consider the impacts of climate change on sea-level rise (Hermans et al. 2020), changes in oxygen concentration (Wakelin et al. 2020, Galli et al., 2024), changes in circulation patterns (Holt et al. 2018), and the impacts of climate change on primary production (Holt et al. 2016). Those have further been used to consider impacts on higher trophic levels and human uses of the marine environment, such as fisheries, aquaculture, conservation, as well as on the full marine ecosystem (Palmer et al.

2021, Queirós et al. 2021, Queiros et al. 2023, du Pontavice et al. 2023, Townhill et al. 2023).

The key aim of this study is to understand how future marine heatwaves will change in northwest Europe under a high emissions greenhouse gas scenario. We separately assess the future evolution of marine heatwaves at the surface and the seafloor and furthermore disaggregate changes seasonally. This assessment will provide a more nuanced and complete understanding of future marine heatwaves on shelf seas.

**2 Methods**

**2.1 Physical models and climate change scenario**

Future marine heatwaves on the northwest European Shelf were projected using the high emissions climate change scenario Representative Concentration Pathway 8.5 (RCP 8.5) (van Vuuren et al. 2011) using an existing ensemble of dynamically downscaled global climate models. Each global climate model used for the downscaling (Holt et al. 2022) is from the

CMIP5 project (Taylor et al. 2012) and was downscaled using the Atlantic Margin Model at 7 km resolution based on the Nucleus of a European Model of the Ocean (NEMO). A detailed explanation of this model can be found in O'Dea et al. (2017). The model employs an equal-angle quadrilateral C-grid mesh with a spatial resolution of 1/15° latitude (7.4 km) and 1/9° longitude (ranging from 5.2 km to 9.4 km), and it features 51 terrain-following vertical levels. The model's geographic domain covers 20°W to 13°E and 40°N to 65°N. For the purposes of heatwave calculations, we defined the northwest

European Shelf as the region shallower than 200 m, and also considered the Norwegian Trench.

In the downscaling approach described by Holt et al. (2022) the surface and lateral boundary conditions, such as air temperature, and initial conditions are taken from the global climate model without bias correction. Due to the poor representation of the boundary connecting the North and the Baltic Seas, a present-day climatology (Gräwe et al. 2015) was used to define the boundary connecting them. This downscaling approach enables the large-scale atmospheric changes projected by the global climate model to transfer to the fine scale changes projected by the regional model. IThe ensemble of



dynamically downscaled projections is fully described in Holt et al. (2022), who produced a 7-member ensemble of future projections that reasonably represent seasonal stratification and present-day temperature. This 7-member ensemble included two pairs of models which used global models which were either closely related to each other or were identical models of differing spatial resolution. We therefore reduced the ensemble to 5 members to ensure full independence. The global

climate models included in the downscaling were as follows: CanESM2 (Swart et al., 2019), CNRM-CM5 (Voldoire et al. 2013), GFDL-ESM2M (Dunne et al. 2020), IPSL-CM5A-MR (Boucher et al., 2019) and MIROC-ESM (Watanabe et al. 2011). In each case, the downscaled projections were run from 1980 to 2099.

## 2.2 Calculation of marine heatwaves

The aim of this study is to provide ecologically relevant projections of future marine heatwaves. In general, the ability of
ecosystems to adapt to warming is thought to be slower than the current and projected future rates of warming (Oliver et al. 2021). We therefore chose to calculate marine heatwaves using a fixed climatology and not using a rolling baseline.

Marine heatwaves were defined and classified based on the methodology of Hobday et al. (2016). Heatwaves were defined as periods where temperatures were above historically high levels for a prolonged period. Historically "high" temperatures were defined as the 90th percentile of temperature within an 11-day window centred on each day of the year for a pre-
defined baseline period. For this study we defined the baseline period to be the years 1990-2009, which is typically viewed as a large enough climatological period for assessing projected changes (Kajtar et al., 2022). The baseline period includes a period of historical forcing 1990-2004 and 5 years of scenario forcing, i.e., assumed and not actual atmospheric $CO_2$ emissions etc. However, the scenario assumptions are identical in each model run, making the model runs directly comparable.

Furthermore, we classified heatwaves as Moderate, Strong, Severe or Extreme using the approach of Hobday et al. (2018). In this approach heatwaves are classified based on multiples of the differences between the 90th percentile and the climatological average temperature on each day of the year. A heatwave of maximum temperature which represents the equivalent of a multiple of 1-2 times this difference is classified as moderate, strong is 2-3, severe is 3-4 and extreme is >4. So, for example, if the 90th percentile historically represented a temperature anomaly of 1 C, a heatwave with a maximum
anomaly of 1.5 C would be classified as moderate, whereas a heatwave of maximum anomaly of 3.5 C would be classified as severe. As with the historical percentiles, we calculated the historical average temperature using an 11-day window for the baseline period.

The ability of the models to reproduce historical (1990-2009) variation in surface temperature during summer was assessed by comparing the anomaly of the historical heatwave temperature threshold in the model with observations. The anomaly is
defined as the daily threshold – the average temperature on each day. Historical daily sea surface (at a spatial resolution of 0.05 by 0.05°) was acquired from the Operational Sea Surface Temperature and Sea Ice Analysis system (OSTIA) (Good et al., 2020) provided by the United Kingdom Met Office. This was downloaded from the Copernicus Marine Environment Monitoring Service (CMEMS: marine.copernicus.eu; https://doi.org/10.48670/moi-00168).



While the impact of winter heatwaves has been observed in the region in a small number of instances (e.g. Atkinson et al.,
2020), negative impacts of marine heatwaves have almost exclusively been assessed during the warm seasons, during the
months of spring and summer (e.g., Bass et al. 2023). Marine heatwave projections should therefore be disaggregated
seasonally (Li et al. 2023, Song et al. 2023) to separate warm season (e.g., summer) heatwaves, which have well established
negative consequences, from the rest of the year, when they have poorly understood consequences. We therefore separated
heatwaves by season.


## 3 Results

In line with expectations, all models project an increase in sea temperatures across the northwest European shelf by the end
of the 21st Century (Figure 1). Reflecting the fact that climate models have broad climate sensitivities (Scafetta, 2022),
projected temperature increases vary across the ensemble. The lowest average projected increase in surface temperature
between 1990-2009 and 2080-2099 is 1.34°C in GFDL-ESM2M and the highest is 4.14 °C in the MIROC-ESM model, with
an average of 2.55 °C across the 5-member ensemble. Notably, across all models, there is a higher level of warming in the
seasonally stratified surface waters, which is notable in the northern North Sea (Figure 1), compared with the seafloor. This
reflects how increased thermal stratification (Holt et al., 2022) will result in less warming translating to deeper waters in the
future compared with today. Across the ensemble surface warming during summer is projected to be 0.88 °C higher than at
the seafloor, while differences in warming in winter are negligible (Figure S1).





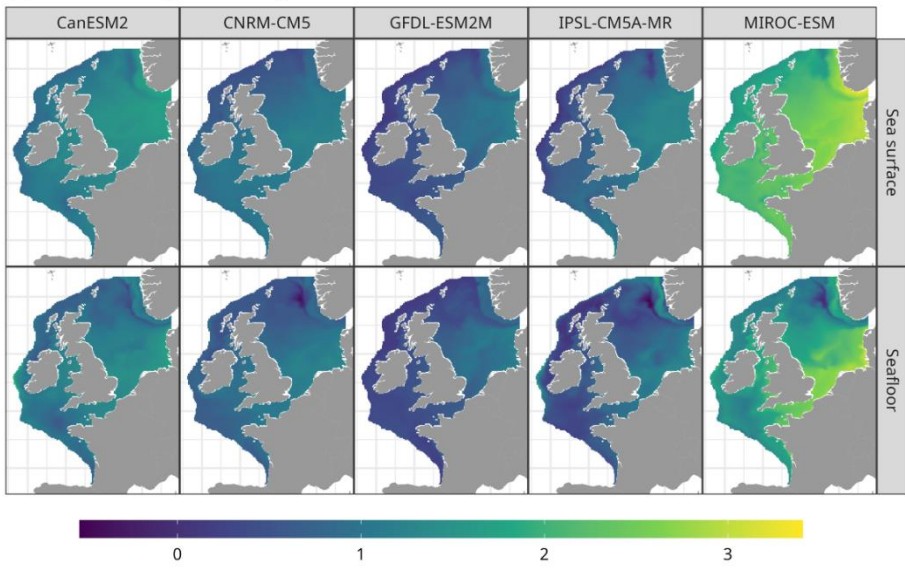

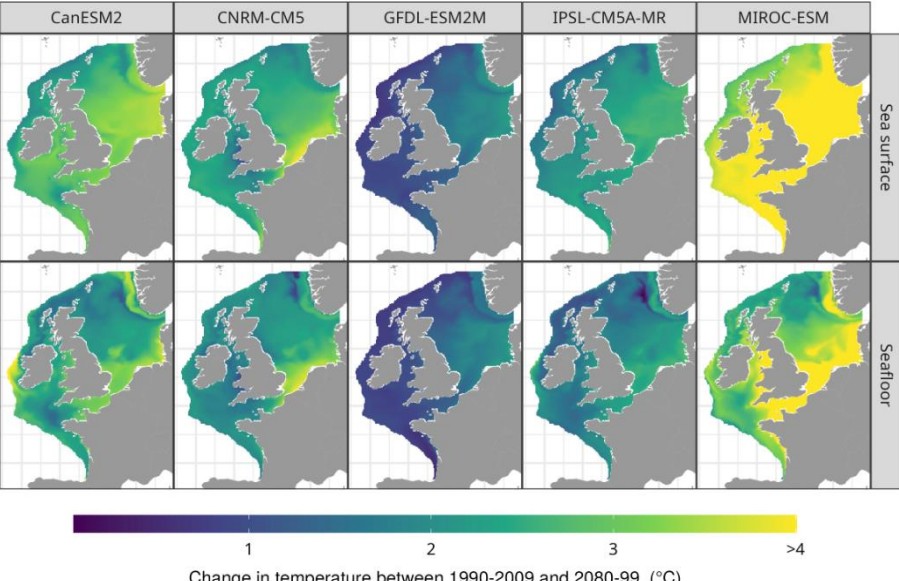

**Figure 1: Projected change in average temperature between a historical baseline period of 1990-2009 and mid-century (2040-59) and end-century (2080-99) periods from five dynamically downscaled climate models under the climate change scenario RCP 8.5. The column names list the global climate model used in the downscaling.**

This large level of ocean warming translates to a large and sometimes rapid increase in annual marine heatwave frequency across the northwest European Shelf across model runs (Figure 2, 3 and 4, table 1). In line with warming levels, the MIROC-





ESM run projects the highest levels of heatwave frequency, with them occurring almost year-round by the end of the 21st Century across much of the shelf at the sea surface. Similarly, the GFDL-ESM2M model projection shows the lowest future frequency of marine heatwaves, but it still projects surface heatwaves to occur approximately 63% of the year at the end of the 21st Century on average across the shelf. The shelf average annual frequency of marine heatwaves at the surface in mid-century ranges from 29.5 -90.5% and from 63.3-99.03% at the end of the century. Similarly, the annual frequency of marine

heatwaves at the seafloor in mid-century ranges from 39.66-96.4% and from 77.22-99.69% at the end of the century.



## (a) Marine heatwaves during 2040-2059

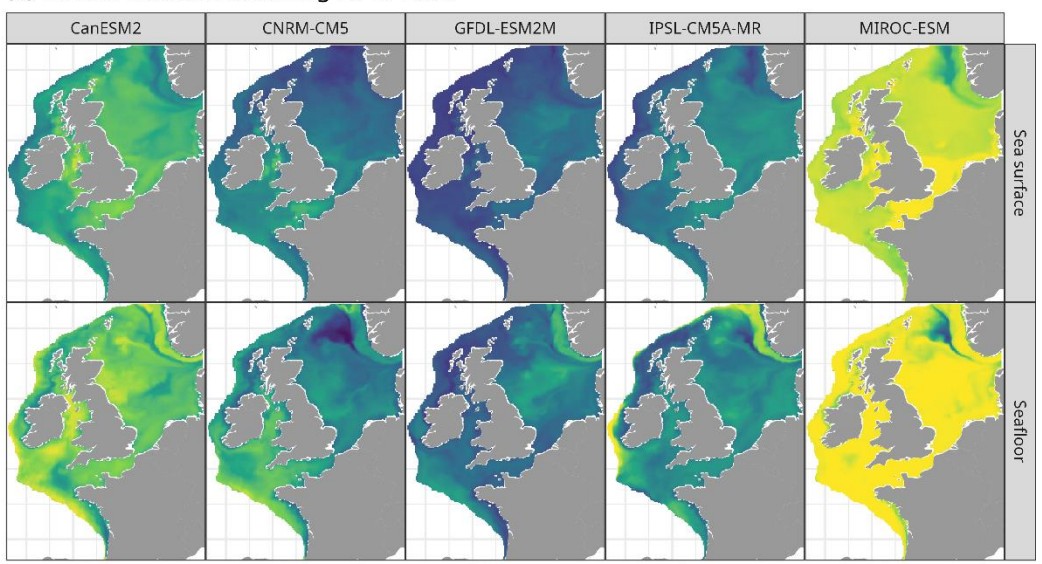

## (b) Marine heatwaves during 2080-2099

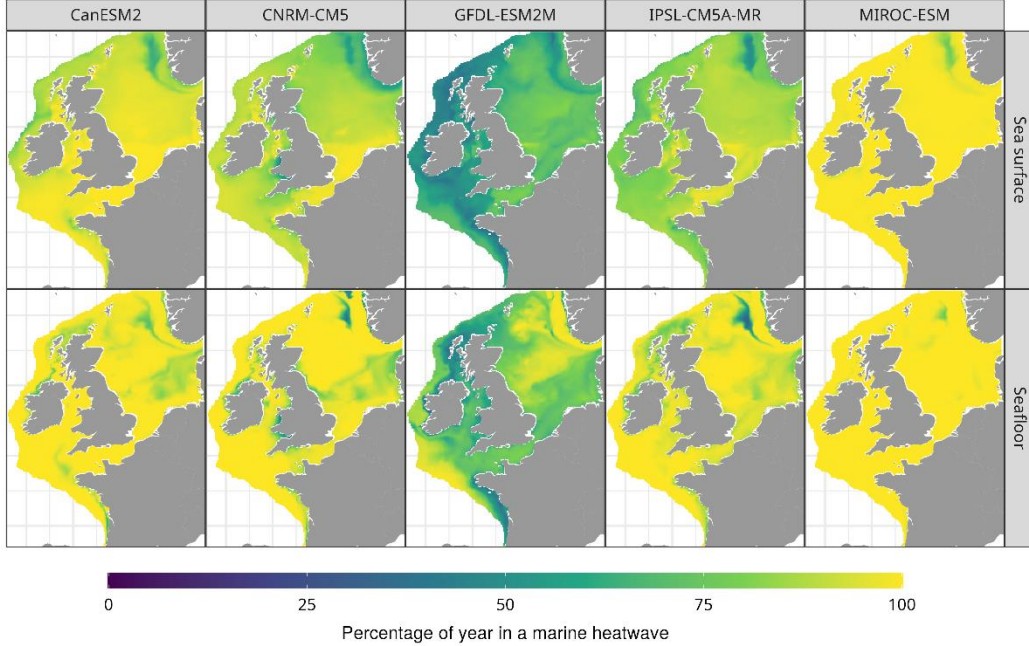

**Figure 2: Projected annual heat waves in mid-century (2040-2059) and end of century (2080-2099), as derived from five dynamically downscaled climate models under RCP 8.5. Heat waves were defined as periods lasting at least 5 days where the daily temperature is greater than the 90th percentile of temperature in the baseline climatological period of 1990-2009.**





**Figure 3: Projected average frequency, using a 20-year rolling average, of marine heatwaves on the North West European Shelf annually and in each season from 1990-2100, using five dynamically downscaled climate models under climate change scenario RCP 8.5.**



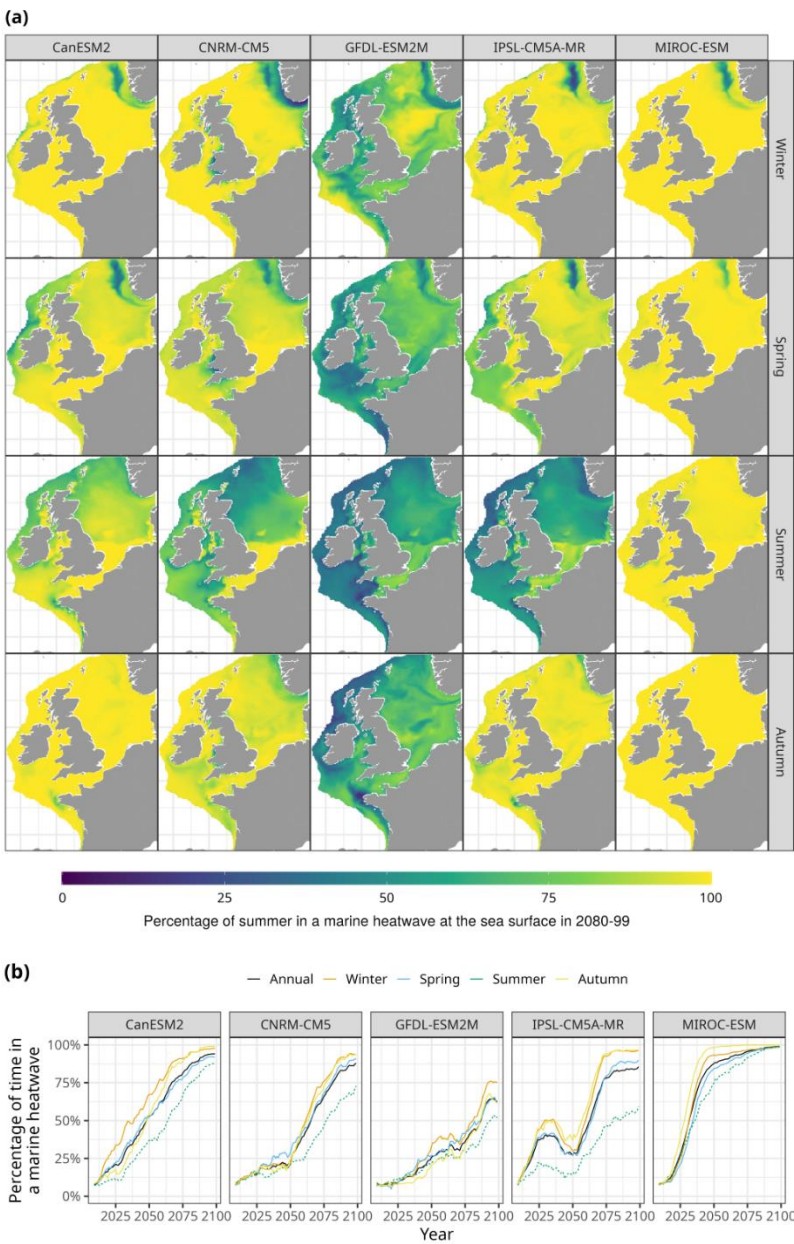

**Figure 4: a Projected average marine heatwave frequency at the sea surface for each season in 2080-2099 using five dynamically downscaled climate models and climate change scenario RCP 8.5. Each column shows the projection from one model. b) Average shelf-wide marine heatwave frequency from 1990-2099 using a 20-year rolling average, for each season and annually. Summer is shown with a dashed line.**



| Surface heatwaves | Model | Period | Annual | Winter | Spring | Summer | Autumn |
|---|---|---|---|---|---|---|---|
| | CNRM-CM5 | 2040-59 | 39.4 | 45.43 | 42.79 | 24.5 | 44.9 |
| | | 2080-99 | 87.33 | 93.84 | 90.57 | 71.1 | 93.8 |
| | CanESM2 | 2040-59 | 62.72 | 76.75 | 61.91 | 43.09 | 69.12 |
| | | 2080-99 | 94.06 | 97.68 | 91.92 | 87.6 | 99.05 |
| | GFDL-ESM2M | 2040-59 | 29.52 | 38.46 | 31.11 | 25.09 | 23.41 |
| | | 2080-99 | 63.03 | 75.33 | 63.1 | 51.7 | 62 |
| | IPSL-CM5A-MR | 2040-59 | 41.84 | 48.67 | 37.31 | 26.59 | 54.81 |
| | | 2080-99 | 84.45 | 96.08 | 88.76 | 56.34 | 96.62 |
| | MIROC-ESM | 2040-59 | 90.59 | 94 | 87.58 | 81.65 | 99.13 |
| | | 2080-99 | 99.03 | 98.4 | 98.94 | 98.8 | 99.99 |
| Seafloor heatwaves | **Model** | **Period** | **Annual** | **Winter** | **Spring** | **Summer** | **Autumn** |
| | CNRM-CM5 | 2040-59 | 51.35 | 48.81 | 53.68 | 52.15 | 50.76 |
| | | 2080-99 | 96.04 | 96.97 | 97.98 | 95.07 | 94.14 |
| | CanESM2 | 2040-59 | 74.47 | 80.63 | 79.11 | 68.62 | 69.55 |
| | | 2080-99 | 97.05 | 99.51 | 99.43 | 94.78 | 94.49 |
| | GFDL-ESM2M | 2040-59 | 39.66 | 40.21 | 44.87 | 41.54 | 32.03 |
| | | 2080-99 | 77.22 | 77.87 | 82.24 | 77.32 | 71.44 |
| | IPSL-CM5A-MR | 2040-59 | 50.66 | 56.41 | 50.98 | 45.3 | 49.96 |
| | | 2080-99 | 95 | 98 | 97.67 | 91.68 | 92.64 |
| | MIROC-ESM | 2040-59 | 96.41 | 97.82 | 99.12 | 95.21 | 93.5 |
| | | 2080-99 | 99.68 | 99.97 | 99.99 | 99.64 | 99.11 |

**Table 1: Projected percentage of time marine heatwaves occur in the future on the northwest European shelf annually and across each season for the middle and end of the 21st Century.**


While the pattern of increased heatwave occurrences is evident at both the sea surface and seafloor, the projections show that the increase is expected to be larger at the seafloor. This is reflected by mapped changes (figure 2) and time series of average changes in heatwaves (figure 3). Overall, the model projections show a clear pattern where there are minimal differences in projected heatwaves at the sea surface and seafloor during winter when stratification is negligible, while there are

pronounced differences in summer when waters are stratified (figures 3 and 4). We illustrated these differences by identifying the first 20-year period when, on average, surface waters are in heatwave conditions for at least 50% of the summer. This highlights the magnitude of differences when temperatures will exceed upper thermal thresholds for species. Figure 5 shows that during these 20-year periods summer heatwaves in deep waters are much more frequent than at the surface. The most extreme difference is the projection from IPSL-CM5A-MR, which shows that in 2060-80 that, on average

the sea surface will be in heatwave conditions 50.4% of the summer, but the seafloor will be in heatwave conditions 89.9%



of the summer. Furthermore, these differences are most pronounced in seasonally stratified waters, while permanently mixed waters such as the southern North Sea see smaller differences between the surface and seafloor.

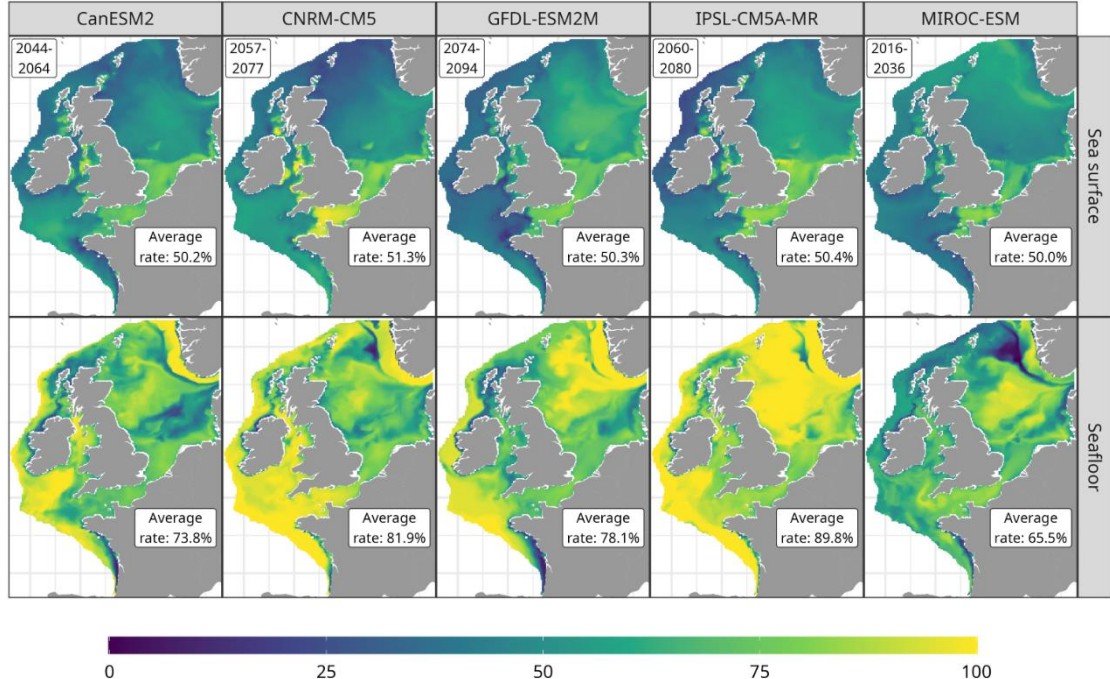

Percentage of summer in heatwave when shelf-wide summer surface heatwaves first occur more than 50% of the time

**Figure 5: Heatwaves at the sea surface and seafloor in the first 20-year period when, on average, heatwaves occur**
**more than 50% of the summer at the surface. The 20-year period shown in the top-left of each panel indicates when**
**the 50% threshold is first exceeded. In the bottom-right of each plot, the average fraction of time the shelf is in a**
**heatwave at the surface or seafloor is shown.**

This difference between heatwaves at the surface and seafloor runs counter to the fact that warming is projected to be higher at the surface than the seafloor. Instead, it should be viewed as consequence of the lower natural inter-annual variability of temperature at the seafloor. This is shown in Figure 6, which shows how much warmer, on average, heatwave temperature thresholds are than the average temperature in the models and observational data. In seasonally stratified waters, the 90th percentile threshold is much lower relative to average conditions at the seafloor than at the surface. Consequently, it takes lower warming levels to translate into more frequent heat waves. Critically, we also find that the ability of the models to





represent inter-annual variability in surface temperature varies, as measured by comparing the temperature anomaly of the 90th percentile threshold with an observational figure (figure 6). Specific geographic features of individual projections may therefore be partly due to biases in the representation of inter-annual variability. For example, in the northern North Sea, the GFDL-ESM2M has lower inter-annual variability in summer than is observed, which partly translates into higher future surface heatwaves than would otherwise be expected. These differences highlight the importance of an ensemble approach to

projecting heatwaves.

### (a) Anomaly of summer heatwave temperature thresholds

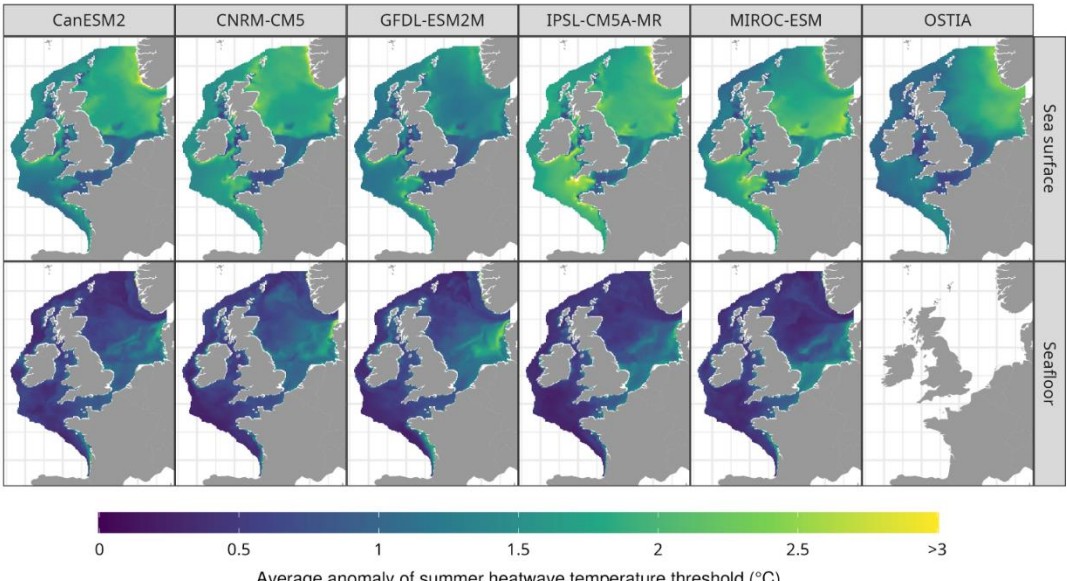

### (b) Bias of summer sea surface heatwave temperature threshold

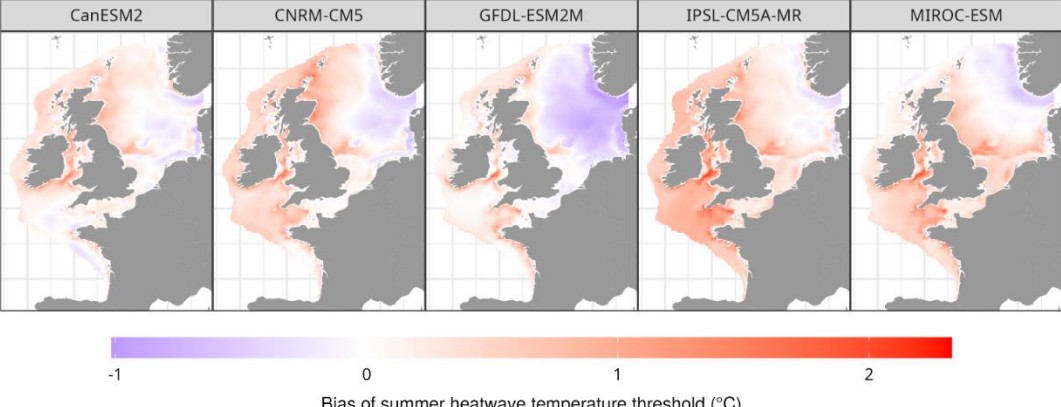

**Figure 6: Average anomaly of climatological heatwave temperature threshold during summer, as derived from five dynamically downscaled climate models. The threshold is defined as the 90th percentile of climatological temperature on each day in the baseline period of 1990-2009 using an 11-day window centered on each day. The anomaly is**





**defined as the difference between the threshold and the average temperature on each day. Surface layer represents the sea surface cell in model output, while bottom layer represents the model cell closest to the seafloor. Column labels name the downscaled global climate model used, with OSTIA representing the thresholds derived from satellite sea surface temperature. Bias represents the difference between model and satellite threshold, i.e., a negative bias implies the model's temperature variability during summer is too low.**

These differences in inter-annual variability have further consequences. Summer temperatures vary more than in the rest of the year (Fig S2). As a consequence, we find that heatwaves increase in frequency more slowly during summer than in the rest of the year (Figures 3 and 4. Table 1). This is most pronounced at the surface, where across the ensemble, sea surface heatwaves occur on average 40.2% of the time during summer and 52.8% of the time annually in 2040-59 and 73.1% of the time during summer and 85.6% of the time annually in 2080-98. The impact of these differences is even more pronounced

when the severity of heatwaves is considered (Figure 7), where extreme sea surface heatwaves are much less likely in summer than they are on average. Notably, in four of the model projections, extreme surface heatwaves are less than 50% as frequent in summer than in the rest of the year at the end of the century. This indicates that marine heatwaves need to be categorized at a seasonal level to accurately assess their impact.







**Figure 7: Projected average frequency of marine heat waves by category at the sea surface, averaged across the northwest European Shelf, using five dynamically downscaled climate models under RCP 8.5. Heat waves are stacked by colour, i.e., the total frequency of heat waves is shown by the top line. The left column shows the average frequency of heatwaves each year, and the right column shows the frequency during summer. A 20-year rolling average is shown, with the output aligned to the final year in each 20-year period.**



## 4 Discussion

Up until now, most research projecting future marine heatwaves in the world's oceans has focused on the sea surface and changes are typically not disaggregated seasonally. We have shown that this may lead to an incomplete picture on shelf seas.

On the northwest European shelf, future seafloor heatwaves are expected to grow at a significantly faster rate than at the sea surface, while summer heatwaves are expected to be less frequent than in the rest of the year. This has nuanced implications for marine ecosystems.

Our study highlights the complex relationship between rising temperatures and marine heatwaves, and how focusing on temperature change alone may mask many nuances. We find that across the northwest European Shelf, seafloor temperatures

are projected to rise at a lower rate than at the surface due to increasing thermal stratification (Holt et al., 2022). However, a lower level of warming on the seafloor may not translate to lower levels of future marine heatwaves. In fact, due to the lower variability of seafloor temperatures, we find that lower levels of warming are required to shift them to heatwave conditions compared to surface waters, and therefore that it is more likely that seafloor habitats will experience more frequent heatwave events than the surface ocean in the region. This indicates that assessing future heatwaves purely based on surface waters is

likely to significantly underestimate future heatwave impacts on many benthic communities in shelf waters. This finding has important implications for seafloor communities and processes in the region, especially for species where population dynamics and lower genetic plasticity resulting from millions of years of more stable thermal conditions may limit the ability of communities to adapt (Somero et al. 2010, Hiddink et al. 2014).

A critical finding in this study is that summer heatwaves in northwest Europe are projected to be relatively less frequent and

potentially significantly less severe than during the rest of the year. This is not because summer warming is expected to be less severe per se, but because of how heatwave calculations are affected by present-day temperature variability. This indicates that assessing the impacts of marine heatwaves purely based on annual changes may overestimate their impacts, on the assumption that marine heatwaves primarily have a significant negative impact in summer. To date, assessments of the ecological impacts of marine heatwaves have largely focused on warm summer months (Smith et al. 2022) when

temperatures reach their highest, there is still much sparser data about the potential ecological heatwaves in cooler months (e.g., Shanks et al. 2019). It is therefore critical that research establish whether heatwave impacts largely reside in summer months when temperatures are highest.

Increasing attention has been placed on defining marine heatwaves in ways meaningful to ecosystem management, with some arguing that a shifting baseline should supplement the fixed baseline approach (Amaya et al. 2023), the latter being

carried out here. A shifting baseline approach is likely to be of significant value for pelagic ecosystems, which as suggested by marine ecosystem models, can reorganize very rapidly to climate change. Pelagic ecosystems are likely to be adapted to a climate resembling a shifting baseline. However, it is unclear if the shifting baseline approach is sensible for all ecological systems, and particularly benthic systems. Seabed benthic species may have poor ability to track more suitable habitat in response to extreme weather events, due to organisms having low dispersal potential or being adapted to historically stabler,

deeper water environments (Somero 2010). This may partially explain why seabed invertebrates in the North Sea have been found to have poor ability to shift range distributions in response to seabed temperature changes (Hiddink et al. 2014).



Identifying regions that are more resilient to climate change is an important task in providing scientific projections of relevance to marine management (Queiros et al. 2021). Notably, our multi-model ensemble shows that some seafloor regions, such as northwest Ireland and the area west of the Norwegian Trench in the North Sea is less impacted by future

marine heatwaves than most of the northwest European shelf. This finding indicates that some regions could be reasonably resilient to future heatwaves.

Our study highlights the importance of looking at heatwave impacts on waters deeper than the sea surface. The general conclusion of the study that seafloor heatwaves are likely to be greater in future than at the surface is likely to be transfer to most shelf seas due to the common existence of seasonal stratification. However, future research should consider the

magnitude of the difference in other regions. Furthermore, our conclusion sits within the context of a growing body of research indicating that the subsurface heatwaves have grown more rapidly than the surface (Sun et al. 2023, He et al. 2023). This study provides the most comprehensive assessment to date of future heatwaves on the northwest European shelf. However, a number of aspects require future attention. For ecosystems and species that can reorganize quickly in response to climate change, projections using a shifting baseline (Giménez et al. 2024) may be more informative. Secondly, many

species have critical temperatures, which can either trigger key life cycle events (Wilson et al. 2024) or represent thermal limits, above which there is rapid mortality or tissue damage (Savva et al. 2018). Projecting future heatwaves, based on known thermal thresholds for key species is therefore the next step in understanding how marine ecosystems will change in future.

*Data Availability*. The projections for sea surface and seafloor temperatures are openly available: https://gws-access.jasmin.ac.uk/public/recicle/.

*Author contributions*. RJW, AMQ and YA conceived the research. RJW wrote the paper, with input and revisions from all authors.

*Competing interests*. The authors decline that they have no competing interests.

*Acknowledgements*. This project has received funding from the European Union's Horizon 2020 research and innovation programme under grant agreement no. 820989 (project COMFORT, Our common future ocean in the Earth system – quantifying coupled cycles of carbon, oxygen, and nutrients for determining and achieving safe operating spaces with respect to tipping points) and from the NERC projects RECICLE (Resolving Climate Impacts on shelf and CoastaL sea Ecosystems; NE/M004120/1 and NE/M003477/2), FOCUS (Future states Of the global Coastal ocean: Understanding for Solutions; 310 NE/X006271/1) CLASS (Climate Linked Atlantic Sector Science; NE/R015953/1), and MSPACE (Marine Spatial Planning Addressing Climate Effects, NE/V016725/1).





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
