# Peer review of "Seafloor marine heatwaves outpace surface events in future on the northwest European shelf"

_EGUsphere, 2024_

## Author Comment (AC1)

I very much appreciate the opportunity to review this manuscript. My major comment is about the threshold used to define marine heatwaves, which is the "90th percentile of climatological temperature on each day in the baseline period of 1990-2009 using an 11-day window centered on each day". This is a fixed baseline method. For future projections with distinct long-term warming trends, authors may consider using shifting baseline as well, e.g., Amaya et al. (2023), to separate the long-term warming signal, especially for the sea surface temperature.  Amaya, D. J., Jacox, M. G., Fewings, M. R., Saba, V. S., Stuecker, M. F., Rykaczewski, R. R., ... & Powell, B. S. (2023). Marine heatwaves need clear definitions so coastal communities can adapt. Nature, 616(7955), 29-32.

We agree that a shifting baseline is of increasing interest as shown by the paper linked and the recent Smith et al. (2025) paper

https://www.sciencedirect.com/science/article/pii/S0079661124002106

For this paper we choose to focus purely on a fixed baseline for the purposes of clarity. It would be challenging to include shifting baselines without ending up with a paper that is very cumbersome. Secondly, this would result in a paper with an unclear focus, as it would move from being about the surface vs. Seafloor to being a paper that some might see as contributing to the fixed vs. Shifting baseline debate. Finally, and perhaps, most important, it is not quite clear if the shifting baseline approach can be applied to the surface and seafloor in the same way. The shifting baseline approach assumes that marine ecosystems have "adapted", for example via intraspecific changes or changes in species composition to the temperature conditions over a recent baseline period, say the last 30 years. However, benthic species are understood to adapt more slowly to climate change, primarily due to their slower movement and often their slower growth. A shifting heatwave baseline should therefore be defined differently for the sea surface and seafloor. However, to date no work has been carried out to identify how this can be done, and carrying out this work is beyond this paper as it is largely an applied, not conceptual paper.

Other comments are listed below:

The model ability to reproduce historical SST variation has been assessed, but its ability to reproduce bottom temperature variation is not included or mentioned at all.

It is probably not possible for the paper to evaluate the ability of models to reproduce variation in seabed temperature due to the lack of high resolution bottom temperature data. The use of reanalysis datasets, such as the CMEMS NWS product which combines models and in-situ data via data assimilation: (https://data.marine.copernicus.eu/product/NWSHELF_MULTIYEAR_PHY_004_009/description) could get you part of the way to an answer. However, problematically this reanalysis

uses NEMO as its ocean model. This is the same model as we have used, and therefore structural biases will carry over.

However, the ability of the model to reproduce bottom temperature variation is likely to track that of the surface. Holt et al. 2022 showed that the model can reproduce seasonal stratification successfully, and it should therefore reproduce the patterns of temperature variation at the sea floor in a similar way to the surface.

Figure 4a colorbar label suggests "percentage of summer in a marine heatwave at the sea surface in 2080-99" but this figure includes all four seasons, which is confusing.

There was a mistake in the figure legend. This is now corrected.

Figure 4b and Figure 3 seem to provide duplicate info about marine heatwave frequency at the sea surface for annual and each season. Figure 4b seems to just plot those lines together instead of separating them in each panel.

Figure 4b has now been removed.

Minor comments:

Line 100: lThe -> The

This typo has been corrected.

Line 103: "which used global models which" -> "which used global models that"

This is now corrected.

Lines 124-125: degree Celsius symbol error

The degree symbol is now fixed.

Figure S1 caption: "north west" -> "northwest"

This has been corrected.

Figure 2 caption: "heat waves" -> "heatwaves"

The term "heat wave" has been changed to heatwave throughout, for consistency.

Figure 3 caption: "North West" -> "Northwest"

This has been corrected.

---

## Author Comment (AC2)

Reviewer comments on "Seafloor marine heatwaves outpace surface events in future on the northwest European shelf" by Wilson et al.

The authors analyze the occurrences and characteristics of marine heatwaves (MHW) on the Shelf Northwest in a warming climate in an ensemble of downscaled climate model simulations. They follow the definition of Hobday et al. and apply a fixed baseline approach. The authors concentrate on MHWs at the surface and the seafloor and contrast the respective MHW occurrences as well as give more details on seasonal differences.

The article is well written and helps to deepen our understanding of MHW conditions in this study region and by shedding light on the differences between sea surface and sea floor and between the seasons. The majority of MHW literature so far focuses on the surface and annual MHW values.

Below are some comments and questions for consideration.

Introduction:

I would like to see a short discussion on the choice of the emission scenario used here for the future prediction, RCP8.5. I realize that, especially in high-resolution, regional modelling it's often the first or only one to be produced, but I think it is important to qualify that this is the high-end scenario and maybe if quantify what degree of warming could be expected in other scenarios in this region. The degree of warming determines the frequency, duration and severity of MHWs.

We agree with the reviewer that the reliance on RCP 8.5 in regional assessments is problematic (https://www.nature.com/articles/d41586-020-00177-3). However, the key aim of the paper is to understand whether future heatwaves at the surface and seabed will differ. We show that across all projections, future seafloor heatwaves will outpace those at the surface. This is almost certain to occur with other scenarios. Given the lack of divergence between RCP 8.5 and the other scenarios until mid-century, we further expect that the general conclusions about the differences between the sea surface and sea bottom should be robust to choice of scenario for much of this century.

We have now provided a couple of sentences to justify the use of a single scenario.

I would also like to see a short discussion on the choice of MHW analysis based on fixed or moving baselines. It is discussed later in the discussion section. I would like to see it earlier and read why the authors chose to apply the fixed baseline approach.

The opening section of 2.2 explained the choice of a fixed baseline. However, we have now extended this to provide a more complete justification.

For this study region I would like to see a more pronounced introduction to the characteristics and causes of MHW in mid-latitudes and shallow water. A lot of MHW literature is based on open ocean and or (sub-) tropical regions. Here we have more short-lived events and high-frequency triggers for onset and decline of MHWs. Stratification plays a role too. A short description of the study region in terms of stratification would be helpful as well.

A couple of sentences have now been added to the introduction about regional drivers. A sentence has also been added to show that the complexity of regional drivers on MHW on the northwest European shelf necessitates the use of high-resolution regional models.

Methods:

Would be nice to read or see a very brief summary of the model evaluation in terms of temperature and most importantly stratification.

Evaluation of stratification was carried out by Holt et al. 2022. We have now added a sentence in the methods to refer readers to this paper.

Temperature is difficult to evaluate within the paper, as the regional model is being driven by global climate models. However, previous work has evaluated the ability of the AMM7 configuration of NEMO to reproduce regional temperature patterns. We have now cited that literature in the methods.

Results:

Lines 145-147: SST changes are described. Absolute warming at the seafloor, min, max, mean?

We feel that the discussion of temperature changes is sufficient. Differences in warming at the surface and seafloor are discussed later in the paragraph. The temperature changes are largely contextual information for the results on marine heatwaves, so we do not believe that highly detailed information is necessary. Changes in maximum and minimum temperature are probably not necessary due to the use of year-round heatwaves, as they would need to be calculated for every day of the year.

Line 195 ff: Does a change of timing and length of stratification play a role?

Changing stratification likely plays a role due to the expected increase in thermal stratification this century (Holt et al. 2022). However, we have not disaggregated the extent of its role in projected changes.

Discussion:

Line 266: Does the degree of variability change in the projections?

This is a good question. We have yet to assess changes in variability. This would be non-trivial due to the need to distinguish between sources of variability, as part of any change in variability will be partly due to the warming trend. We therefore did not assess this issue but are considering doing so in future.

Line 284-286: Because they are deeper or next to a deep region?

This is probably partly due to changing advection patterns moving colder deeper water towards these regions. Though disaggregating this is beyond the scope of the paper.

Do lateral flows at the seafloor play a role?

Cooling occurs in isolated regions primarily due to changes in water flows. This shows up largely on the edge of the Norwegian Trench. Changes in flow in the western Norwegian Trench were previous discussed by Holt et al. (2018).

https://agupubs.onlinelibrary.wiley.com/doi/full/10.1029/2018GL078878

Figures:

Figure 1: Colormap / color limits: is there cooling (dT < 0°C)? Hard to tell. Why would that be the case? Have the same color limit for a and b.

The colour scale has now been changed to blue-white-red to clearly show that regional cooling can occur. The colour limits have also now been unified.

Figures: I would prefer different colormaps for temperature changes and MHW days respectively.

We have now changed Figure 1 so that it has a blue-red colour scale. This distinguishes clearly between the temperature and heatwave plots.

Figure 2 caption: Unit would be something like MHW days per year in % Or percentage of heatwave conditions?

This figure caption has now been changed to be clearer.

Figure 3,4,7 caption similar. For me, frequency means the number of MHW events.

This figure caption has been improved.

Figure 3: Can you explain the dip in the warming curve in the IPSL model? Unfortunately, it falls into the mid-century period in this analysis...

The IPSL model projects a regional dip in air temperature, which feeds through to the ocean temperatures. We are unaware of published literature on the explanation.

Figure 4b: At the current size the lines are hard to see/distinguish.

Figure 4b has now been removed, as it was partly repeating information from figure 3.

Minor comments:

Throughout:

check for missing ° signs

These should now be corrected.

for better readability consider having 1 or 0 digits for percent numbers in the text and in tables

We agree. The tables have now been changed.

Line 100: typo

This is now corrected.

Line 130: sea surface temperature

This is corrected.

Line 143: Blank space

This has been removed.